The appropriateness of ceftriaxone and metronidazole as empirical therapy in managing complicated intra-abdominal infection—experience from Western Health, Australia

Tan Andrew asktan@gmail.com
Rouse Michael
Kew Natalie
Qin Sharon
La Paglia Domenic
Pham Toan
Department of General Surgery, Western Health , Melbourne , Victoria , Australia
Tulkens Paul
Electronic publication date: 2018 Aug 15
Publication date: 2018
Volume: 6
Electronic Location ID: e5383
Received 2018 Feb 28; Accepted 2018 Jul 16
Copyright: ©2018 Tan et al.
Copyright year: 2018
Copyright holder: Tan et al.
License: This is an open access article distributed under the terms of the Creative Commons Attribution License, which permits unrestricted use, distribution, reproduction and adaptation in any medium and for any purpose provided that it is properly attributed. For attribution, the original author(s), title, publication source (PeerJ) and either DOI or URL of the article must be cited.
License URL: https://creativecommons.org/licenses/by/4.0/

Keywords: Intra-abdominal infection, Empirical antibiotics, Ceftriaxone, Metronidazole

Funding: The authors received no funding for this work.

==============================
Purpose

This study aims to assess the microbiological profile, antimicrobial susceptibility and adequacy of intravenous ceftriaxone and metronidazole as empirical therapy for surgical patients presenting with complicated intra-abdominal infection.

Methods

This retrospective audit reviews the microbiological profile and sensitivity of intra-abdominal cultures from adult patients with complicated intra-abdominal infection who presented to the emergency department at Western Health (Melbourne, Australia) between November 2013 and June 2017. Using the hospital’s database, an audit was completed using diagnosis related group (DRG) coded data. Ethics approval has been granted by the Western Health Human Research Ethics Committee. Results are stratified according to surgical conditions (appendicitis, cholecystitis, sigmoid diverticulitis and bowel perforation). The antimicrobial coverage of ceftriaxone and metronidazole is evaluated against these microbial profiles.

Results

A total of 1,412 patients were identified using DRG codes for intra-abdominal infection. All patients with microscopy and sensitivity results were included in the study. Patients without these results were excluded. 162 patients were evaluable. 180 microbiological cultures were performed through surgical intervention or radiologically guided aspiration of the intra-abdominal infection. Single or multiple pathogens were identified in 137 cultures. The most commonly identified pathogens were mixed anaerobes (12.6%), Escherichia coli (E. coli) (12.1%), mixed coliforms (11.6%) and Pseudomonas aeruginosa (7%). Other common pathogens (6% each) included Enterococcus faecalis, Streptococcus anginosus, Vancomycin-resistant Enterococci (VRE) and Extended Spectrum Beta-Lactamases (ESBL) producing E. coli. Organisms isolated in our study are consistent with existing literature. However, a significant proportion of antibiotic resistant organisms was identified in cases of perforated bowel and sigmoid diverticulitis. Broader spectrum antimicrobial therapy should therefore be considered in lieu of ceftriaxone and metronidazole in these cases. Ceftriaxone and metronidazole remain as appropriate empirical therapy for patients who presented with perforated appendicitis and cholecystitis.

Discussion

The empirical regime of ceftriaxone and metronidazole remains appropriate for intra-abdominal infection secondary to appendicitis and cholecystitis. In cases involving perforated small and large bowel, including complicated sigmoid diverticulitis, the judicious use of ceftriaxone and metronidazole is recommended.

Introduction

Intra-abdominal infection (IAI) is a common condition in surgery and is an important cause of morbidity and mortality, despite therapeutic advancements in recent times (Weiss, Steffanie & Lippert, 2007). The Surgical Infection Society (SIS) (Mazuski et al., 2017) outlines the main therapeutic approaches to the management of IAI, which include the expeditious recognition of IAI, early resuscitation of the patient, adequate risk assessment of patient factors, timely and appropriate source control of IAI either via surgical or percutaneous drainage, and finally, the early initiation of appropriate antimicrobial therapy.

IAI can be further classified as complicated or uncomplicated. Uncomplicated IAI is defined as “intramural inflammation of the gastrointestinal tract without anatomic distortion” (Lopez, Kobayashi & Coimbra, 2011b). These infections are typically simple to treat, provided treatment is not inappropriate or delayed. Complicated IAI (cIAI) “extends beyond the hollow viscus of origin into the peritoneal space and is either associated with abscess formation or peritonitis” (Solomkin et al., 2010).

In clinical practice, empirical antimicrobial therapy is typically commenced once a patient is recognised to have an IAI. Delaying antimicrobial treatment has been associated with poor outcomes (Kumar et al., 2006; Rhodes et al., 2017). As such, empirical antimicrobial therapy typically precedes surgical or radiological intervention. Common practice for the acute management of complicated IAI at our health service, Western Health (Melbourne, Australia), is to commence empirical antimicrobial treatment with intravenous ceftriaxone and metronidazole (“empirical therapy”), with subsequent adjustment as guided by microbiological profile and sensitivity tests. The use of ceftriaxone and metronidazole as combination therapy is consistent with SIS’s recommended therapy for low-risk adults and children (Mazuski et al., 2017).

However, in a proportion of cases, there is limited response to this empirical therapy, leading to clinical deterioration and subsequent invasive procedures that may have been avoided. In this context, two clinical questions arise: 1. Whether empirical therapy using ceftriaxone and metronidazole provides sufficient antimicrobial coverage in IAI overall, and 2. whether the microbiological profile of IAI differs according to diagnosis (e.g., appendicitis versus diverticulitis), thereby requiring different antimicrobial therapies tailored to the disease. The rise of antibiotic resistant microorganisms worldwide in IAI further compounds the selection of appropriate empirical antimicrobial treatment (Sartelli et al., 2015). Therefore, the changing epidemiology of the microorganisms in IAI requires active surveillance through microbiological profiling and susceptibility testing.

The purpose of our study is to assess the microbiological profile, antimicrobial susceptibility, and the adequacy of intravenous ceftriaxone and metronidazole as empirical antimicrobial therapy for patients who present to our health service with complicated IAI.

Methods

This retrospective study reviews the microbiological profile and sensitivity of intra-abdominal cultures from adult patients with general surgical conditions who presented with complicated IAI to the emergency department at Western Health (Melbourne, Australia) between November 2013 and June 2017. Ethics approval (with waiver of patients’ consent) was granted by the Western Health Low Risk Human Research Ethics Approval (ref: QA2017.64). Data was collected for patients with intra-abdominal collections which were cultured through surgical intervention or radiologically guided percutaneous aspiration. Patients with complicated IAI were identified through the hospital database using Diagnosis Related Group (DRG) and International Classification of Diseases-Version 10-Australian Modification (ICD-10 AM) codes. A total of 1,412 records were reviewed. Microbiological and susceptibility profiles were collected for all IAI cases and then stratified according to four common general surgical conditions for comparative analysis: perforated appendicitis, sigmoid diverticulitis, cholecystitis and bowel perforation. The antimicrobial coverage of ceftriaxone and metronidazole was then evaluated against the microbiological and sensitivity profiles collected from the intra-abdominal cultures.

Results

Of 1,412 patients identified by DRG codes, 162 patients met the inclusion criteria. The mean age of all patients was 55 years. One hundred and eighty separate microbiological cultures were obtained from either surgical intervention or radiologically guided aspiration of the intra-abdominal infection. Single or multiple pathogens were identified in 137 cultures (76.1%). The most commonly identified pathogens were mixed anaerobes (12.6%), Escherichia coli (12.1%), mixed coliforms (11.6%) and Pseudomonas aeruginosa (7%) (Fig. 1). Other common pathogens (approximately 6%) include mixed enterococci, Enterococcus faecalis, Streptococcus anginosus, Vancomycin-resistant Enterococci (VRE), and Extended Spectrum Beta-Lactamases (ESBL) producing E. coli. Based on this distribution, we estimate that the use of our empirical therapy would provide antimicrobial coverage for 56.5% of all cases.

Figure 1 Top 15 organisms as identified from cultures.

Acute appendicitis

There were 49 patients who presented with acute appendicitis and localised or diffused peritonitis, corresponding to the Sunshine Appendicitis Grading Score (SAGS) (Reid et al., 2017) of 2 and above. There were 30 men and 19 women. The mean age was 45 years, with a mean length of stay of eight days. 43 patients were commenced on intravenous ceftriaxone and metronidazole; six were commenced on either tazocin or a combination therapy of amoxicillin, gentamicin and metronidazole.

Positive cultures were identified in 39 (79.6%) patients. The most commonly identified microorganisms were mixed coliforms (20.4%), Streptococci spp. (18.4%), E. coli (14.3%) and mixed anaerobes (14.3%). It is important to note however that there were three cases of P. aeruginosa and two cases of ESBL-producing E. coli—all are resistant to our empirical therapy, thereby requiring changes to the treatment regime. All five cases were associated with perforated gangrenous appendicitis (SAGS 4).

Forty-one patients had an uneventful recovery and were discharged. Interestingly, eight patients presented with re-collection after appendectomy. With the exception of one, all cases were related to either gangrenous appendicitis or suppuratives appendicitis with perforation. Six of these patients were commenced on ceftriaxone and metronidazole, with tazocin and the combination of amoxicillin and metronidazole accounting for the remaining two patients. Of these eight patients, four have no significant past medical history. Of the remaining four, two are smokers, and the other two have type 2 diabetes mellitus (T2DM). For these patients, no culture was initially performed during appendectomy. All eight patients subsequently underwent radiologically guided percutaneous drainage. Seven of the eight patients returned positive cultures. Mixed coliforms, E. coli and mixed anaerobes were most common, which were susceptible to our empirical therapy. However, there was one positive identification each of Pseudomonas aeruginosa and ESBL-producing E. coli which were resistant to this combination therapy. The antibiotic regimen was changed accordingly. These patients were subsequently discharged with no further sequelae.

Acute cholecystitis

There were nine patients (six males, three females) who presented with acute cholecystitis with peri-cholecystic collection. The gallbladder was perforated in all patients. The mean age was 68 years and the mean length of stay was 13.4 days. Reflective of the higher mean age in this group, these patients have multiple co-morbidities, including chronic conditions such as T2DM, hypertension and hypercholesterolaemia. Ceftriaxone and metronidazole were empirically commenced for five patients, tazocin for two patients, ceftriaxone only for one patient, and the last patient received treatment with cephazolin and metronidazole.

Of the nine cultures performed, there were five (55.6%) positive cultures. Two cases of E. coli, one each of K. pneumoniae, Strep. anginosus and mixed enteric flora were identified. These microorganisms are typically vulnerable to our empirical therapy, and eight patients were subsequently discharged.

A 68 year old male patient (with a past medical history of T2DM, atrial fibrillation, hypertension and psoriasis) re-presented with a recollection. At his first presentation, he was commenced on ceftriaxone only, and the initial culture taken during cholecystectomy was positive for E. coli, which is typically sensitive to ceftriaxone. Culture taken from percutaneous drainage of the recollection was negative.

Sigmoid diverticulitis

There were 29 patients who presented with sigmoid diverticulitis with intra-abdominal collection. There were 15 women and 14 men. The mean age was 55 years and the mean length of stay was 17 days. These patients were categorised according to Hinchey’s (Hinchey, Schaal & Richard, 1978) classification, relying on intraoperative findings at the first instance or radiological findings if there was no surgical intervention. The classifications are as follows:

Hinchey classifications:

• Grade I: three patients

• Grade II: 13 patients

• Grade III: six patients

• Grade IV: seven patients

Twenty-six patients were empirically commenced on ceftriaxone and metronidazole; two were commenced on amoxicillin and metronidazole, while the remaining patient was commenced on amoxicillin, gentamicin and metronidazole. Eighteen patients underwent surgery (14 Hartmann’s procedure, two anterior resections and two washouts). The remaining 11 patients underwent radiologically guided percutaneous drainage.

There were 26 (89.7%) positive cultures. The most common microorganisms were mixed anaerobes (22.2%), E. coli (14.3%), E. faecalis (11.1%), mixed coliforms (11.4%), and mixed enteric flora (8.6%). ESBL-producing E. coli and P. aeruginosa account for 8.6% and 5.6% respectively—these pathogens, together with E. faecalis, are resistant to our empirical therapy. Accordingly, there is significant antimicrobial resistance to our therapy in our sample. The antibiotic treatment was modified accordingly. Twenty-nine patients had an uneventful recovery and were subsequently discharged.

There were four patients who re-presented to the emergency department with either one or two intra-abdominal re-collections post surgical or percutaneous drainage. Cultures taken from the first presentation identified microorganisms that were susceptible to the empirical therapy for three patients; cultures for the last patient grew P. aeruginosa at the first presentation and he was started on ciprofloxacin prior to discharge. Three patients subsequently underwent laparotomy and washout with either Hartmann’s procedure or anterior resection; one patient had the recollection percutaneously drained. From these recollections, there were five cultures in total of which four (80%) were positive. One case each of P. aeruginosa, ESBL-producing E. coli, VRE and E. coli were identified. With the exception of E. coli, these pathogens were resistant to our empirical therapy. These antibiotic resistant pathogens were identified in patients with Hinchey III (ESBL-producing E. coli) and Hinchey IV (P. aeruginosa and VRE) sigmoid diverticulitis. Looking at this patient group, the four patients with positive cultures were older (ranging from 68 to 71 years old), and with a range of significant co-morbidities including previous uncomplicated diverticulitis (conservatively managed), T2DM, HTN, active smoking status and ischaemic heart disease.

Bowel perforation

There were 31 patients who presented with bowel perforations and associated IAI, exclusive of appendicitis or diverticulitis. There were 18 males and 13 females. The mean age was 55 years, with a mean length of stay of 23 days. 58% of perforations occurred in the large bowel while the small bowel accounted for the other 42%. The most common causes of bowel perforations were attributed to colorectal cancer (eight cases), trauma (five cases) or bowel obstruction (five cases). Other causes include bowel perforations secondary to Crohn’s disease, ulcer, volvulus, radiation enteritis, and ischaemia.

Nineteen patients were empirically commenced on ceftriaxone and metronidazole; five on tazocin, one on meropenem and the others on either cephazolin or amoxicillin and metronidazole. Twenty-eight patients subsequently underwent laparotomy, while three patients underwent laparoscopic surgery.

Of the 31 cultures performed, 23 were positive (74%). The most common microorganisms identified were Candida spp. (16.2%), mixed anaerobes (11%), E. faecalis (11%), ESBL-producing E. coli (8%), and mixed coliforms (8%). VRE and P. aeruginosa accounted for 5.4% each. Antibiotic resistant organisms were identified from both small and large bowels in equal proportion. Antimicrobial treatment for these patients were modified in accordance with the culture results. Given the significant presence of antibiotic resistant microorganisms in bowel perforations, the judicious use of ceftriaxone and metronidazole as empirical antimicrobial therapy is recommended. Twenty-seven patients were subsequently discharged with no significant sequelae.

However four patients (three males, one female) re-presented to the hospital with re-collections. Three are below the age of 60, while one patient was 77 years old. They first presented for bowel perforation secondary to Crohn’s, volvulus, trauma (foreign body) and obstruction. All four patients underwent laparotomy at their first presentation, with three having had a complex surgical course with multiple subsequent laparotomies and a prolonged length of stay. Two patients subsequently underwent percutaneous drainage of the re-collections; one underwent multiple laparotomies while the remaining patient underwent a repeat laparotomy and a subsequent percutaneous drainage procedure. There were six cultures taken from these re-collections, of which four were positive (67%). Two cases of Klebsiella oxytoca, one of Candida tropicalis, and one mixed enterococci were identified.

Figure 2 summarises the organisms identified by surgical condition and their susceptibility to our empirical therapy of ceftriaxone and metronidazole.

Figure 2 Organisms as identified by surgical condition and their susceptibility to empirical therapy of ceftriaxone and metronidazole.

Discussion

The microbiological profiles from our results are typical of enteric flora, comprising predominantly of coliforms and anaerobes, including E. coli. This is the case for patients who presented with acute appendicitis and cholecystitis. However, for cases with sigmoid diverticulitis and bowel perforations, organisms resistant to our empirical therapy of ceftriaxone and metronidazole had been isolated. This includes E. faecalis, ESBL-producing E. coli, VRE and P. aeruginosa. The Tokyo Guidelines have made recommendations on intravenous antimicrobial agents for treatment of IAI—this includes the use of ceftriaxone for lower risk patients and metronidazole as anti-anaerobic agent (Mazuski et al., 2017). The guidelines also recommend the use of broad-spectrum antibiotics such as carbapenem based therapy for higher risk patients (age ⩾ 70 years; malignant disease; significant cardiovascular, hepatic, or renal disease; hypoalbuminemia; generalised peritonitis; delayed initial source control; inability to achieve adequate source control; or suspected infection with resistant pathogens) and to target many ESBL-producing strains of enterobacteriaceae.

Our results from acute appendicitis are consistent with the literature, which shows E. coli and Streptococci spp. to be the most prevalent organisms (García-Marín et al., 2018). Anaerobes such as Bacteroides spp. and Prevotella spp. have also been commonly isolated. These organisms typically respond well to our empirical therapy, as well as to amoxicillin-clavulanic acid. P. aeruginosa is a frequent isolate in different series for appendicitis (6–35%), although ours was slightly lower at 5% (Jeon et al., 2014). In cases of complicated IAI secondary to appendicitis, the SIS recommends obtaining cultures of peritoneal fluid in higher risk patients to identify any resistant or opportunistic pathogens and to analyse epidemiologic data (Mazuski et al., 2017).

In cases of acute cholecystitis, studies have shown that gram-negative Enterobacteriaceae such as E. coli (31–44%), Klebsiella spp. (9–20%) and Enterobacter spp. are commonly isolated from bile cultures. For gram-positive organisms, Enterococcus spp. (3–34%) and Streptococcus spp. (2–10%) are the common isolates (Nitzan et al., 2017; Gomi et al., 2013). Results from our cases are concordant with the literature. Cases of P. aeruginosa (0.5–19%) have been reported, although this has not been isolated in our study. Several empirical antimicrobial options are available for treatment of acute cholecystitis. The Tokyo Guidelines recommend the combination therapy of a cephalosporin such as ceftriaxone, with or without metronidazole, and carbapenem based therapy such as ertapenem (Grade 1 recommendation) (Gomi et al., 2013). The use of ampicillin-sulbactam is not recommended without the addition of an aminoglycoside. The use of ceftriaxone and metronidazole as empirical treatment at our health service for acute cholecystitis is in line with the recommended regime. Additionally, the pathogens isolated from our study are susceptible to our empirical therapy.

The offending pathogens isolated from diverticulitis are typically representative of the colonic flora, as colonic perforations lead to IAI. This includes gram positive and negative as well as anaerobic organisms (predominantly Bacteroides fragilis). The latter outnumber aerobic and facultative organisms on the order of 1,000:1 (Byrnes & Mazuski, 2009). Patients are typically older. Brook et al. evaluated 110 patients with peritonitis related to diverticulitis over a 15-year period (Byrnes & Mazuski, 2009), and found that three-quarters of the specimens were polymicrobial. E. coli was isolated in 71% of patients, and gram-positive organisms (predominantly Streptococci spp.) were identified in approximately 10–20% of patients. Additionally, B. fragilis was isolated in 50% of patients. Other anaerobes such as Clostridium spp. and Fusobacterium spp. were also seen. The recommended antimicrobial regimen include ceftriaxone and metronidazole for diverticulitis associated IAI (Mazuski et al., 2017; Byrnes & Mazuski, 2009). Our results are generally in accordance with the literature. However, we have also identified a clinically significant population of organisms which are resistant to our empirical therapy. These include ESBL-producing E. coli, E. faecalis and P. aeruginosa. Accordingly, the use of broader-spectrum antimicrobial therapy for higher risk patients should be considered (Mazuski et al., 2017).

Jang et al. (2015) examined the microbiological profile of 419 patients who presented with secondary peritonitis due to perforation of hollow viscus, excluding perforated appendicitis and cholecystitis. Peritoneal cultures were positive in 69% of patients. The most commonly isolated organisms were E. faecium (35.2%), E. coli (20%), Candida albicans (17.2%), and P. aeruginosa (17; 11.7%). Streptococcus spp. and Bacteroides spp. were identified in 7.6% and 6.9% of patients, respectively. Other pertinent findings include, for example, that blood culture positivity was significantly more common in colonic perforation, and that Klebsiella spp was more commonly isolated in cases of small bowel perforation. Additionally, Enterococcus spp and E. coli were more commonly identified in perforation of the lower gastrointestinal tract. More importantly, antibiotic resistant organisms were identified in approximately 10% of patients. These include VRE, ESBL-producing E. coli and K. pneumonia, Methicillin Resistant Staphylococcus Aureus (MRSA), and carbapenem-resistant P. aeruginosa and Acinetobacter baumannii. These antibiotic resistant organisms were more commonly found in perforations of the colon (21.3%), stomach/duodenum (18.8%) and small bowel (17%). Their findings were consistent with previous aetiological studies (Skrupky, Tellor & Mazuski, 2013).

The organisms isolated from cultures in cases of large and small bowel perforation in our study are similar to existing literature, with predominance in Candida spp., anaerobes and E. faecalis. Ceftriaxone as monotherapy is not the antibiotic of choice against Enterococci spp. (Barbara, 2018). Ampicillin is the recommended agent of choice against E.  faecalis, if not using piperacillin-tazobactam or imipenem-cilastatin (Mazuski et al., 2017). Additionally, the presence of ceftriaxone-resistant organisms such as VRE, P. aeruginosa and ESBL-producing E. coli, totalling 18.8% of positive cultures, suggests that ceftriaxone does not provide sufficient antimicrobial coverage in cases of bowel perforation in our study. Broader spectrum agents such as piperacillin-tazobactam or carbapenems may be better suited in this context. The significant presence of these antibiotic resistant organisms as well as yeast species reiterate the importance of primary source control and microbiological profiling for targeted therapy in order to minimise treatment failure.

The primary aim of our pilot epidemiological study is to examine the microbial profile of patients who presented to our health service with complicated IAI, and to assess if ceftriaxone and metronidazole remain appropriate as empirical antimicrobial therapy for these patients. It has several limitations which can be addressed in future research: (1) it is a retrospective study with a limited sample size, (2) the relationship between patient factors (age, co-morbidities) and antibiotic resistant microorganisms is not sufficiently explored, (3) the role of the timing of both the commencement of antimicrobial therapy and subsequent intervention is not examined.

Conclusion

In our population, the empirical regime of ceftriaxone and metronidazole remains appropriate for IAI secondary to perforated appendicitis and cholecystitis. In cases involving perforated bowel or complicated sigmoid diverticulitis, given the significant presence of organisms resistant to our empirical therapy, the judicious use of ceftriaxone and metronidazole is recommended.

Supplemental Information

Data S1 Formatted and de-identified dataset

Click here for additional data file.

Additional Information and Declarations

Competing Interests

Author Contributions

Human Ethics

Data Availability

The authors declare there are no competing interests.

Andrew Tan conceived and designed the experiments, performed the experiments, analyzed the data, prepared figures and/or tables, authored or reviewed drafts of the paper, approved the final draft.

Michael Rouse and Sharon Qin performed the experiments, analyzed the data, contributed reagents/materials/analysis tools, authored or reviewed drafts of the paper, approved the final draft.

Natalie Kew performed the experiments, analyzed the data, contributed reagents/materials/analysis tools.

Domenic La Paglia conceived and designed the experiments, contributed reagents/materials/analysis tools, authored or reviewed drafts of the paper, approved the final draft.

Toan Pham conceived and designed the experiments, analyzed the data, authored or reviewed drafts of the paper, approved the final draft.

The following information was supplied relating to ethical approvals (i.e., approving body and any reference numbers):

Ethics approval (with waiver of patients’ consent) was granted by the Western Health Low Risk Human Research Ethics Approval (ref: QA2017.64).

The following information was supplied regarding data availability:

The formatted and de-identified dataset is provided in a Supplemental File.

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
