# Peer review of "The appropriateness of ceftriaxone and metronidazole as empirical therapy in managing complicated intra-abdominal infection—experience from Western Health, Australia"

_PeerJ, doi:10.7717/peerj.5383_

## Round 0.1 · original submission · Major Revisions

As you will see, your contribution triggered somewhat contradictory recommendations (minor vs major revision). Based on my own analysis and taking the remarks of the reviewers into consideration, I wish to give you a chance to have your paper accepted if you can answer appropriately to the remarks raised. Thus, your rebuttal and detailed explanation for what has been modified or not in the paper will be essential to reach a final decision if you submit a revised version.

Reviewer 1 ·

Basic reporting

This retrospective study has a clear aim - to identify the pathogens involved in common surgical conditions that lead to intra-abdominal abscesses. They also nicely showed whether their recommended antibiotic use was appropriate treatment for those same organisms due to their sensitivities.

The article is generally clear and unambiguous; there are a few grammatical errors which require addressing and have been listed below.

The Introduction is concise and sets the scene well. More recent literature references could be added (please see below).

The references are generally up to date and appropriate

The figures are good and are both relevant and well labelled.

Experimental design

The authors have nicely presented a single centre experience in the microorganism presentation for surgical intra abdominal abscesses. There are some larger studies in the literature but they tend to only cover specific conditions rather than the whole spectrum which the authors have done.

The research question is well defined and the results are simple but clear and relevant to common practice.

There are weaknesses in this study; it is reliant on DRG coding and when they subdivided the conditions causing intra abdominal abscesses, some of the groups were of smaller numbers (eg cholecystitis group n= 9)

Validity of the findings

Although this is a retrospective study, the search they performed is sound and they have produced a descriptive analysis of their results.

The data is robust.

The conclusion is supported by the results which they identified is correct - they have illustrated that in more complex perforations, there is a higher likelihood of the presence of resistant organisms and therefore require consideration of an alternative antibiotic regime.

Additional comments

The strengths of this study are that it provides a summary of organisms and their sensitivities in a range of surgical conditions from a single treating institution. I think that the take home message is clear and is translatable to clinical practice.

I would support this articles acceptance, but with the following conditions are addressed / answered:-

1. Patients who had resistant organisms - were there any underlying risk factors for these (previous recurrent admissions for antibiotic therapy for the same condition)
2. Did all patients who were sensitive to Ceftriaxone and Metronidazole respond to this therapy or were there patients who had to have second line antibiotic therapy

3. Line 73 - there is more recent literature evidence supporting that the delay of sepsis control is associated with poorer outcomes.
4. Line 84 - grammatical change - In this context two clinical questions arise ;- 1. Whether empirical therapy using ceftriaxone and metronidazole provides sufficient antimicrobial coverage in IAI overall and 2………
5. Line 209 - to correct sentence IAI - this includes the "use of" ceftriaxone

6. Line 224 - What is the definition of "higher risk patients"…

·

Basic reporting

the authors don't stress between un complicated and complicated IAI
and the relative involving bacteria and manage and microbiological sampling
the epidemiological and/or local incidence of ESBL and MBL bacteria in the described setting is not showed

Experimental design

the aim and scope should be include the difference above cited: describe the epidemiology of setting, the kind of infections complicated and uncomplicated
characteristics of population comorbidities, previous hospitalization ect

Validity of the findings

the manuscript showed the efforts of authors to find an empirical treatment of cip antibiotic treatment despite the field is very difficult to manage

Additional comments

I appreciate the efforts of the authoris but the matter it is so complex that deserves to characterize better the surgical aspect, intervention in emergency or election, local epidemiology and to comment with help of the literature if this empirical therapy can be applied only in your hospital or is an option for setting with the same epidemiological microbiological isolates

---

## Round 0.2 · accepted · Accept

Thank you for making the necessary corrections. This has been most useful.

Reviewer 1 ·

Basic reporting

.

Experimental design

.

Validity of the findings

.

Additional comments

As per my previous review - I was generally happy and feel a further review not necessary on my side.